# In-hospital outcomes and 30-day readmission rates among ischemic and hemorrhagic stroke patients with delirium

**Farhaan S. Vahidy** [1]*, **Arvind B. Bambhroliya**[1], **Jennifer R. Meeks**[1], **Omar Rahman**[2], **E. Wesley Ely**[3,4], **Arjen J. C. Slooter**[5], **Jon E. Tyson**[6], **Charles C. Miller**[6], **Louise D. McCullough**[1], **Sean I. Savitz**[1], **Babar Khan**[2,7]

1 The Institute of Stroke and Cerebrovascular Diseases and The Department of Neurology, McGovern Medical School at University of Texas Health Science Center, Houston, Texas, United States of America, 2 Division of Pulmonary/Critical Care, Indiana University School of Medicine, Indianapolis, Indiana, United States of America, 3 Critical Illness, Brain Dysfunction, Survivorship (CIBS) Center, Vanderbilt University Medical Center, Department of Medicine, Nashville, Tennessee, United States of America, 4 Tennessee Valley Veteran's Affairs Geriatric Research Education Clinical Center (GRECC), Nashville, TN, United States of America, 5 Department of Intensive Care Medicine and Brain Center Rudolf Magnus, University Medical Center Utrecht, Utrecht University, Utrecht, The Netherlands, 6 Center for Clinical Research and Evidence Based Medicine, McGovern Medical School at the University of Texas Health Science Center, Houston, Texas, United States of America, 7 Indiana University Center for Aging Research, Regenstrief Institute, Inc, Indiana University Center for Health Innovation and Implementation Science; Indiana Clinical and Translational Sciences Institute, Indianapolis, Indiana, United States of America

* Farhaan.Vahidy@uth.tmc.edu

**Data Availability Statement:** Data analyzed in the study are publicly available for purchase from the Central Distributor of the Healthcare Cost Utilization Project (HCUP) of the US Agency for Healthcare

## Abstract

### Objective

Delirium is associated with poor outcomes among critically ill patients. However, it is not well characterized among patients with ischemic or hemorrhagic stroke (IS and HS). We provide the population-level frequency of in-hospital delirium and assess its association with in-hospital outcomes and with 30-day readmission among IS and HS patients.

### Methods

We analyzed Nationwide in-hospital and readmission data for years 2010–2015 and identified stroke patients using ICD-9 codes. Delirium was identified using validated algorithms. Outcomes were in-hospital mortality, length of stay, unfavorable discharge disposition, and 30-day readmission. We used survey design logistic regression methods to provide national estimates of proportions and 95% confidence intervals (CI) for delirium, and odds ratios (OR) for association between delirium and poor outcomes.

### Results

We identified 3,107,437 stroke discharges of whom 7.45% were coded to have delirium. This proportion significantly increased between 2010 (6.3%) and 2015 (8.7%) (aOR, 95% CI: 1.04, 1.03–1.05). Delirium proportion was higher among HS patients (ICH: 10.0%, SAH: 9.8%) as compared to IS patients (7.0%). Delirious stroke patients had higher in-hospital

Research and Quality for researchers with online Data Use Agreements and completion of the HCUP Data Use Agreement Training (URL: https://www.distributor.hcup-us.ahrq.gov; email: HCUPDistributor@AHRQ.gov). The authors used the Nationwide Readmission Database (NRD) of the HCUP for the study for years 2010-2015. Data were obtained through following steps: completion of the web-based HCUP Data Use Agreement (DUA) Training, registration for an account with the online HCUP Central Distributor, completion of the profile using the DUA Training certification code and completion date, submission of an application for purchase of the HCUP NRD data with online acknowledgment of applicable HCUP DUAs, and submission of the payments. Data were delivered to the authors via secure digital download. Detailed information regarding each step is available at https://www.distributor.hcup-us.ahrq.gov. Description of data elements for the HCUP NRD is available at https://www.hcup-us.ahrq.gov/db/nation/nrd/nrddde.jsp. None of the authors of the present study are affiliated with the HCUP or collected any of the data on behalf of the HCUP or have special access privileges in accessing the HCUP. Data were purchased publicly from the HCUP. All interested researchers can access the data through HCUP in a similar manner through completing and submitting an application for purchasing HCUP Databases at the HCUP Central Distributor (https://www.distributor.hcup-us.ahrq.gov).

**Funding:** We acknowledge the infrastructural support provided by the Biostatistics/Epidemiology/Research Design (BERD) component of the Center for Clinical and Translational Sciences (CCTS) for this project. CCTS is mainly funded by a grant (UL1 TR000371) from the National Center for Advancing Translational Sciences (NCATS), awarded to University of Texas Health Science Center at Houston.

**Competing interests:** The authors have declared that no competing interests exist.

mortality (12.3% vs. 7.8%), longer in-hospital stay (11.6 days vs. 7.3 days) and a significantly greater adjusted risk of 30-day-readmission (16.7%) as compared to those without delirium (12.2%) (aRR, 95% CI: 1.13, 1.11–1.15). Upon readmission, patients with delirium at initial admission continued to have a longer length of stay (7.7 days vs. 6.6 days) and a higher in-hospital mortality (9.3% vs. 6.4%).

## Conclusion

Delirium identified through claims data in stroke patients is independently associated with poor in-hospital outcomes both at index admission and readmission. Identification and management of delirium among stroke patients provides an opportunity to improve outcomes.

## Introduction

Delirium is characterized by an acute disturbance of attention and awareness.[1] Delirium is prevalent among critically ill patients, predisposes to hospital-acquired complications, leads to prolonged intensive care unit (ICU) and hospital stays, higher health care utilization, and increased mortality.[2–5] The presence of ICU delirium is also associated with long-term cognitive impairment and dementia.[6–8]

The majority of the literature on ICU delirium comprises studies that exclude neuro-critical care populations. This is understandable given the difficulties in implementing commonly used delirium screening tools in patients with neurologic catastrophes such as ischemic or hemorrhagic stroke (IS and HS). Based on limited evidence, the frequency of delirium among IS and HS patients reported from single center studies ranges between 12% and 43%.[9, 10] However, the current literature neither provides a comprehensive view of population-based delirium frequency among various types of stroke patients, nor does it shed light on the relationship between delirium and other relevant outcomes such as mortality, length of stay, discharge disposition and early hospital readmissions in neuro-critically ill patients.

We conducted this analysis to provide nationally representative estimates of delirium frequency, as documented in administrative databases, for patients with IS and HS and assess a proportional change in these estimates over a 6-year contemporary time period between 2010 and 2015. We also provide estimates for poor in-hospital outcomes of higher mortality, longer length of stay, and unfavorable discharge disposition as well as 30-day readmission rates for IS and HS patients who experienced delirium as compared to those who did not experience delirium.

## Methods

### Data source

We utilized the Nationwide Readmission Database (NRD) of the HCUP for years 2010 to 2015. It comprises a nationally representative, weighted probability sample of approximately 36 million discharges annually from hospitals of geographically dispersed states in the US (27 states included in 2015). The non-weighted sample represents approximately 50% of total US hospitalizations. Using unique patient-linkage information in the NRD, patients can be tracked for readmissions across hospitals within a state. Statistical analyses were completed by FV and AB, as per data use agreement guidelines of HUCP. Use of de-identified publicly available data did not warrant an institutional review of this study.

## Cohort selection and primary exposure

We used the International Classification of Diseases, Ninth Revision (ICD-9) codes and identified patients with primary discharge diagnosis of IS [ICD-9 433.x1, 434.x1, 436] ICH [ICD-9 431], and sub-arachnoid hemorrhage (SAH) [ICD-9 430]. Delirium was identified using published algorithms for administrative claims databases.[11] These algorithms have high specificity (99%) and positive predictive value (91%) when compared to clinically relevant and validated methods for diagnosis of delirium, such as the confusion assessment method (CAM) and CAM-ICU (S1 Table).[12] We excluded patients <18 years old, and those HS patients who had concurrent diagnoses of head trauma, thus limiting the HS population to primary ICH and SAH. We also excluded patients who were discharged in last quarter of 2015 due to transition of administrative coding to ICD-10.

## Study outcomes

The main study outcomes were in-hospital mortality, length of stay, discharge disposition (categorized as home including home health, transfer to a health care facility, and other) and all cause 30-day hospital readmission. Among the readmitted patients, we also assessed in-hospital mortality, length of stay, and frequency of subsequent delirium. For 30-day readmission outcomes, we identified each unique patient using verified patient-linkage information. For this cohort, the first admission that met the inclusion criteria was considered as the *index admission*. The 30-day readmission was defined as any readmission within and including the 30[th] day post *index admission* discharge. To avoid misclassification of inter-hospital transfers as possible readmissions, we excluded patients who were readmitted on the same day as that of discharge from index admission. We further excluded patients who were discharged within last month of each year to allow adequate follow-up time to observe a 30-day readmission event.

## Co-variates

We compared patients with and without delirium across several demographic (age, gender, insurance-status, median household income for patient's ZIP Code) and comorbidity variables (Charlson co-morbidity score, number of chronic conditions, All Patient Refined Diagnosis Related Group (DRG): Severity of Illness, and multiple AHRQ Elixhauser comorbidity measures).[13, 14] We used ICD-9 code to identify cases who received intravenous thrombolytics (99.10), craniotomy (01.24), craniectomy (01.25), invasive mechanical ventilation (96.70), noninvasive mechanical ventilation (93.90), endotracheal tube placement (31.1, 31.2, 31.21, and 31.29), gastric tube placement (43.11), and extra-ventricular drain placement (02.21).

## Statistical analyses

We report proportions and 95% confidence intervals (CI) of IS and HS patients experiencing in-hospital delirium and used multivariable survey design logistic regression methods to provide an estimate of year-wise change in this proportion. Standard errors and variances were calculated as per published methods.[15] We fitted survey design logistic regression models to assess the association of in-hospital delirium with mortality and discharge disposition, and report adjusted odds ratios (aOR) and CI. We fitted survey design negative binomial and modified Poisson models to examine the association of in-hospital delirium with length of stay and 30-day readmission respectively. Adjusted risk ratios (aRR) and CI are reported for these estimates. All models were controlled for demographics, comorbidities, stroke type and treatment intensity variables. All analyses were performed using statistical software (STATA, version

15.2; StataCorp LP). Based on the number of events (252,687 for in-hospital mortality, and 288,223 for readmission) we had greater than 95% power to detect observed effects and satisfy multivariable regression modeling requirements.

## Results

### Characteristics of the study population and frequency of delirium

For the entire period of investigation (Jan 2010 –Sept 2015) we identified 3,107,437 unique stroke discharges among whom 7.45%, CI (7.33–7.57) were coded to have delirium. The proportion of delirium was higher among patients with hemorrhagic stroke (ICH: 10.0%, SAH: 9.8%) as compared to IS patients (7.0%). Fig 1 demonstrates eligible stroke discharges from the NRD, reasons for exclusion, and the proportion of stroke patient discharges with and without delirium by stroke subtype. The mean (SE) age of the cohort was 70.02 (0.07) years, 51.9% were females, 65.6% had Medicare insurance, and 81.4% were hypertensive. Patients older than 65 years had significantly higher likelihood of having delirium as compared to those younger than 65. (aOR: 1.36, CI: 1.32–1.40 for 65–80 years; aOR: 1.88, CI: 1.82–1.94 for 81–90 years). Females had higher likelihood of delirium as compared to males (aOR: 1.02, CI: 1.00–1.04). S2 Table represents descriptive statistics and univariate analysis for demographic, comorbidity, and disease severity variables for the overall cohort, and for patients with and without delirium. The proportion of stroke patients diagnosed as delirium during hospitalization significantly increased between 2010 (6.3%) and 2015 (8.7%) (aOR, CI: 1.04 (1.03–1.05) (Fig 2).

### In-hospital outcomes among stroke patients with delirium

Stroke patients with delirium had significantly higher in-hospital mortality (12.3% vs. 7.8%, aOR: 1.36, 95% CI: 1.31–1.41) and longer length of stay (11.6 vs. 7.3 days; aRR: 1.24, CI: 1.23–1.25) as compared to stroke patients without delirium. The majority of stroke patients without delirium went home with or without home health (57.1%) whereas the majority of patients with delirium were transferred to other health care facilities (51.9%; aOR for transfer versus home: 1.79, CI: 1.75–1.83). Table 1 reports results of in-hospital outcomes for stroke patients by delirium status. We further analyzed in-hospital outcomes separately for three stroke subtypes (IS, ICH, and SAH). Among patients with IS (constituting 84.3% of the overall cohort), delirium was associated with poor outcomes. However, a lower proportion of HS patients (ICH and SAH) with delirium died during hospitalization as compared to non-delirious HS patients. This seemingly 'protective' effect observed among HS patients remained statistically significant after adjustment in multivariable models for both ICH (aOR, CI: 0.83, 0.78–0.88) and SAH (aOR, CI: 0.75, 0.69–0.83) patients. However, longer length of stay and lower likelihood of home discharge among HS patients with delirium was consistent with findings in the overall cohort. S3A–S3C Table shows results of in-hospital outcomes for three stroke subtypes by delirium status. For HS patients, we further investigated the association between in-hospital delirium and mortality by strata of APDRG mortality risk. Based on this stratified analysis, the association between delirium and death was significantly higher among the low risk patients, however among the higher risk patients delirium was associated with lower proportion of death.

### 30-day readmission and post-readmission in-hospital outcomes

Stroke patients with delirium had a significantly greater adjusted risk of 30-day-readmission (16.7%) as compared to those without delirium (12.2%) (aRR, CI: 1.13 (1.11–1.15). Upon

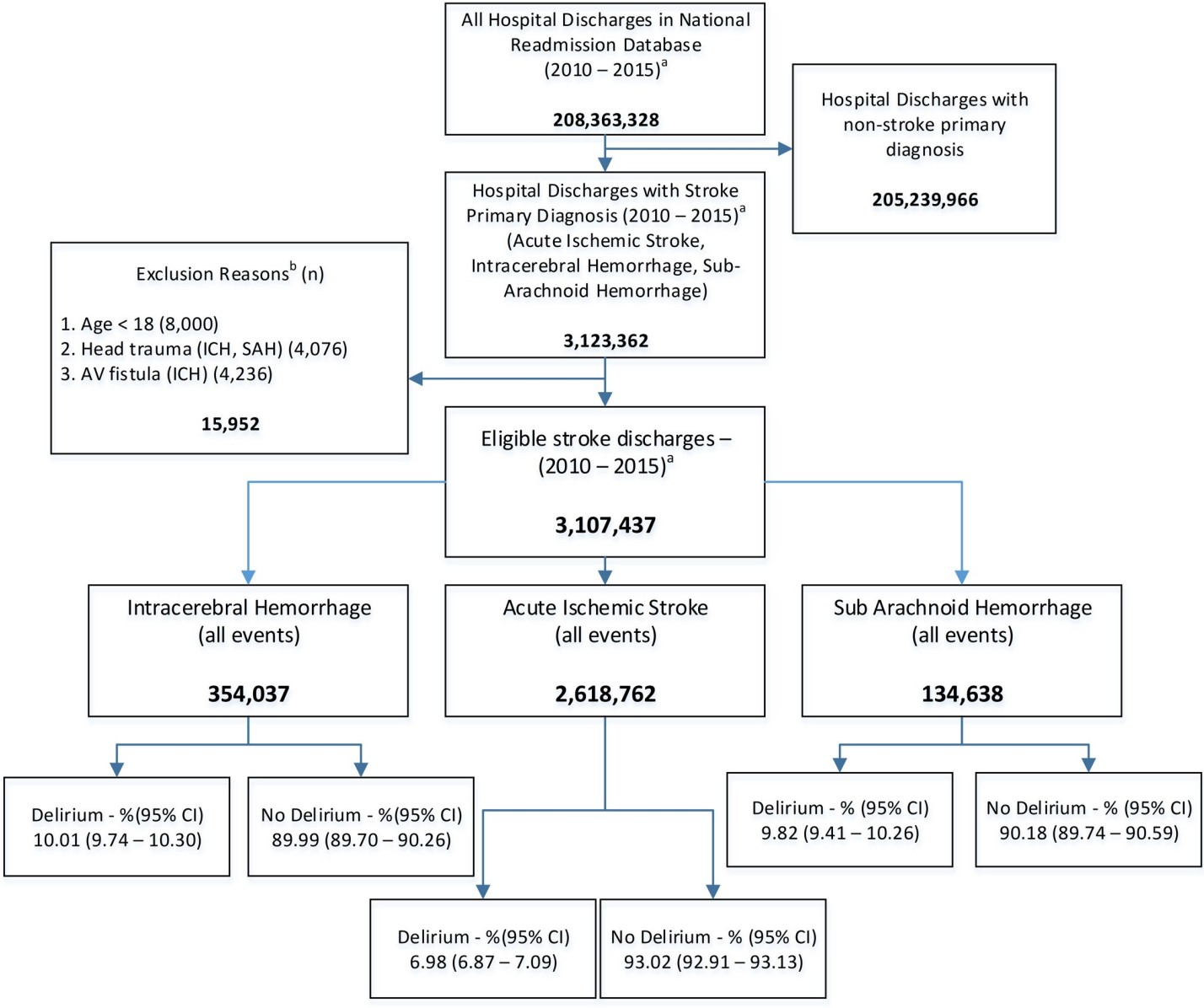

**Fig 1. Eligible stroke discharges, reasons for exclusion, and proportion (95% confidence intervals) of stroke patient discharges with and without delirium between January 1, 2010 and September 30, 2015 in the National Readmission Database.** Proportions and 95% confidence intervals for initial, excluded, eligible, and analyzed population of ischemic and hemorrhagic stroke patients along with proportion and 95% confidence interval for frequency of delirium observed for different stroke subtypes for the entire duration of analyses (2010–2015). The listed reasons for non-inclusion are not mutually exclusive.

readmission, stroke patients with delirium at index admission continued to have significantly higher in-hospital mortality (9.3% vs. 6.4%; aOR: 1.32, CI: 1.21–1.44), and longer length of stay (7.7 vs. 6.6 days; aRR: 1.08, CI: 1.05–1.11). The majority of readmitted patients without delirium at index admission went home (50.5%) whereas the majority of readmitted patients with delirium at index admission were transferred to other health care facilities (56.2%; aOR, CI for transfer versus home: 1.68, 1.58–1.79). Table 2 reports 30-day outcomes for stroke patients by delirium status at index admission. Upon further analyzing 30-day outcomes by stroke subtype, IS patients with delirium had significantly higher risk of 30-day readmission, and higher in-hospital mortality, and longer length of stay upon readmission. For ICH

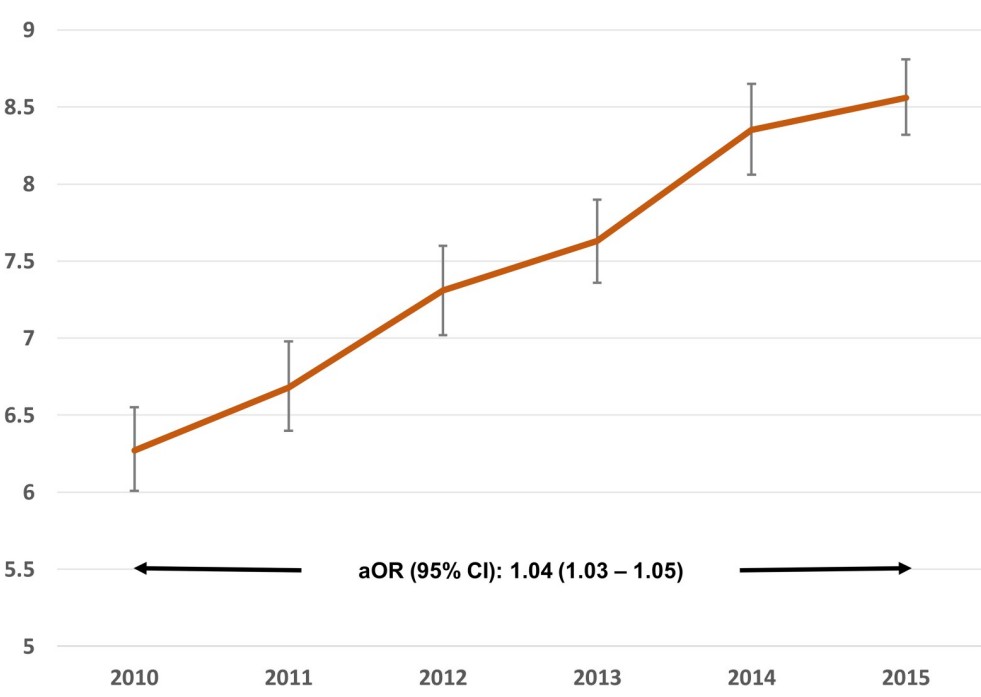

**Fig 2. Proportion of stroke patients coded as delirium per year of analysis. Error bars indicate 95% confidence interval of the proportion.** Year-wise change in proportion of ischemic and hemorrhagic stroke patients diagnosed and coded as having in-hospital delirium from the Nationwide Readmission Database between 2010 and 2015. The reported odds ratio and 95% confidence interval obtained from survey design logistic regression model control for patient demographic, comorbidity, and disease severity factors.

**Table 1. In-hospital outcomes for stroke patient discharges with and without delirium.**

| In-Hospital Outcomes | Total (n = 3,107,437) | No Delirium (n = 2,875,938) | Delirium (n = 231,500) | aOR / aRR (95% CI) |
|---|---|---|---|---|
| Died, %(95% CI) | 8.14 (8.02–8.25) | 7.80 (7.68–7.92) | 12.28 (11.96–12.60) | 1.36 (1.31–1.41)* |
| Length of Stay, mean(SE) | 7.57 (0.04) | 7.25 (0.04) | 11.62 (0.09) | 1.24 (1.23–1.25)† |
| Discharge Disposition, % (95% CI) | | | | |
| Home incl. Home with Home Health | 55.44 (55.10–55.78) | 57.09 (56.74–57.44) | 34.99 (34.50–35.48) | Reference |
| Transfer (Hosp /SNF/ICF/Other) | 35.57 (35.25–35.89) | 34.25 (33.93–34.58) | 51.92 (51.43–52.42) | 1.79 (1.75–1.83)* |
| Died | 8.14 (8.02–8.25) | 7.80 (7.68–7.92) | 12.28 (11.96–12.60) | 1.88 (1.81–1.95)* |
| Other | 0.85 (0.82–0.88) | 0.86 (0.83–0.89) | 0.81 (0.74–0.88) | 1.53 (1.41–1.66)* |

All models controlled for: (1) *Demographic Variables*: Age, Sex, Insurance Type, Income Quartile by Zip Code of Residence (2) *Comorbidities*: Charlson Co-morbidity Index, Number of Chronic Diseases, Atrial Fibrillation, Coagulopathies, Hypertension, Peripheral Vascular Disease, Valvular Disease, Diseases of Pulmonary Circulation, Other Neurological Diseases, Depression, Psychiatric Illness, Chronic Lung Disease, Liver Disease, Diabetes Mellitus (with complications), Renal Disease, Electrolyte Imbalance, Anemia, Chronic Blood Loss, Ulcer, Tumor, Obesity, Drug Abuse, Alcohol Abuse (3) *Stroke Type and Treatment Intensity*: Hemicraniectomy / Craniotomy, Extra Ventricular Drain Placement, Gastric Tube, Tracheostomy, Ventilator Support, Intravenous IV tPA, Intra-Arterial Therapy, Stroke Type

\* Estimates obtained from multivariable *logistic regression models (Odds Ratio)*

† Estimates obtained from *negative binomial regression models (Risk Ratio)*

**Table 2. Rate of 30-day readmission, in-hospital outcomes, and frequency of delirium among readmitted stroke patients with and without delirium at index admission.**

| 30-Day Outcomes | Total (n = 2,078,855) | No Delirium (n = 1,942,014) | Delirium (n = 136,841) | aOR / aRR (95% CI) |
|---|---|---|---|---|
| Readmission Rate, % (95% CI) | 12.49 (12.38–12.60) | 12.20 (12.09–12.31) | 16.65 (16.30–17.00) | 1.13 (1.11–1.15)* |
| **Number of readmissions** | **Total (n = 288,223)** | **No Delirium (n = 262,652)** | **Delirium (n = 25,571)** | **aOR / aRR (95% CI)** |
| Died, %(95% CI) | 6.59 (6.41–6.76) | 6.32 (6.15–6.50) | 9.29 (8.66–9.97) | 1.32 (1.21–1.44)† |
| Length of Stay, mean(SE) | 6.66 (0.03) | 6.57 (0.03) | 7.66 (0.10) | 1.08 (1.05–1.11)‡ |
| **Discharge Disposition, % (95% CI)** | | | | |
| Home incl. Home with Home Health | 49.06 (48.59–19.52) | 50.53 (50.04–51.03) | 33.89 (32.81–34.99) | Reference |
| Transfer (Hosp /SNF/ICF/Other) | 43.68 (43.23–44.14) | 42.47 (41.99–42.95) | 56.17 (55.02–57.31) | 1.68 (1.58–1.79)† |
| Died | 6.59 (6.41–6.76) | 6.32 (6.15–6.50) | 9.29 (8.66–9.97) | 1.72 (1.56–1.90)† |
| Other | 0.68 (0.63–0.73) | 0.68 (0.63–0.73) | 0.64 (0.51–0.81) | 1.29 (1.00–1.67)† |
| **Frequency of Delirium upon Readmission, % (95% CI)** | | | | |
| Delirium on readmission | 12.28 (12.03–12.53) | 10.91 (10.67–11.15) | 26.33 (25.38–27.30) | 1.94 (1.86–2.02)† |

All models controlled for: (1) *Demographic Variables*: Age, Sex, Insurance Type, Income Quartile by Zip Code of Residence (2) *Comorbidities*: Charlson Co-morbidity Index, Number of Chronic Diseases, Atrial Fibrillation, Coagulopathies, Hypertension, Peripheral Vascular Disease, Valvular Disease, Diseases of Pulmonary Circulation, Other Neurological Diseases, Depression, Psychiatric Illness, Chronic Lung Disease, Liver Disease, Diabetes Mellitus (with complications), Renal Disease, Electrolyte Imbalance, Anemia, Chronic Blood Loss, Ulcer, Tumor, Obesity, Drug Abuse, Alcohol Abuse (3) *Stroke Type and Treatment Intensity*: Hemicraniectomy / Craniotomy, Extra Ventricular Drain Placement, Gastric Tube, Tracheostomy, Ventilator Support, Intravenous IV tPA, Intra-Arterial Therapy, Stroke Type

* Estimates obtained from modified Poisson model *(Risk Ratio)*

† Estimates obtained from multivariable *logistic regression models (Odds Ratio)*

‡ Estimates obtained from *negative binomial regression models (Risk Ratio)*

patients, delirium at index admission had significantly greater risk of 30-day readmission but was not significantly associated in-hospital mortality and length of stay upon readmission. For SAH patients, delirium at index admission was not significantly associated with either 30-day readmission, or in-hospital mortality and length of stay upon readmission. However, for both hemorrhagic subtypes (ICH and SAH), delirium at index admission had higher likelihood of transfer to other health care facility compared to discharge to home. S4A–S4C Table shows results of 30-day outcomes for three stroke subtypes by delirium status at index admission. Proportion of delirium upon readmission for all stroke patients with delirium at index admission was higher compared to that for patients without delirium (26.3% vs. 10.9%; aOR: 1.94, CI: 1.86–2.02). Similar results were found when the cohorts were stratified based on stroke-types.

## Discussion

In this large national database of over 3 million stroke discharges, we identified cases of in-hospital delirium across a 6-year contemporary period. Frequency of delirium was higher among hemorrhagic as compared to ischemic stroke patients. Overall, delirium in stroke patients was associated with higher in-hospital mortality, longer length of hospital stay, unfavorable

discharge disposition, and a higher risk of 30-day readmissions. Our findings add to the growing body of literature demonstrating that in-hospital delirium is associated with adverse outcomes, and further extend this evidence to ischemic and hemorrhagic stroke patients. To our knowledge, population-based national estimates of delirium in all subtypes of stroke patients have not been previously reported and the impact of delirium on poor outcomes among these neurocritically ill patients has not been comprehensively evaluated.

Utilizing these nationwide administrative data, we found the overall frequency of delirium to be in the range of 7%; which seems to be lower as compared to other single-center studies. [9] We believe that this underestimation is due to the failure to diagnose or code delirium in an administrative database.[16] Prior smaller studies, conducted in single center settings, probably enrolled selected patients from tertiary care ICUs with active delirium monitoring protocols. Our employed algorithm however has very high specificity and positive predictive value (> 90%), hence almost eliminating false positives and making our estimates valid for stroke patients who were definitively diagnosed and coded as having in-hospital delirium.[11] Our data also demonstrate significant increase in frequency of in-hospital delirium among stroke patients. This is likely a reflection of increasing awareness and documentation. As mentioned above, utilization of validated tools with daily screening practices will provide the true prevalence of delirium in this challenging patient population. The feasibility of utilizing validated tools for diagnosis of in-hospital delirium among neurocritically ill patients has recently been reported.[9]

Our data demonstrate independent association of delirium with poor in-hospital outcomes (death, longer length of stay, and unfavorable discharge disposition) among stroke patients. Though it is possible that delirium in itself may be a biological marker of disease severity among stroke patients, our large sample size allowed us to control for several comorbidity and disease severity variables, and the significant independent association of delirium with poor in-hospital outcomes persisted across all our iterative models. Furthermore, presence of delirium during index hospitalization was also independently associated with higher risk of subsequent early readmission. There are no prior reports examining the risk of early readmission associated with delirium among stroke patients. However, presence of delirium has been reported to be associated with longer and repeated institutionalization for other disease cohorts.[17, 18] Even if delirium among the neurocritically ill is a marker of disease severity, our data signify the importance of its recognition and possible amelioration during hospitalization. Particularly because it has been demonstrated among non-stroke patients that a significant proportion of delirium is preventable and there are effective non-pharmacological strategies for its management.[19, 20]

In addition to higher likelihood of readmission among delirious stroke patients, readmitted patients who had delirium during index hospitalization subsequently had higher in-hospital mortality, longer length of hospital stay, and a higher probability of re-experiencing delirium during the readmission episode. We can hypothesize that certain individual characteristics may predispose patients to develop delirium in the first place. This at the outset seems plausible albeit simplistic given that we adjusted for multiple relevant confounders. The other explanation that is difficult to confirm is that delirium perturbs biologic pathways, for example aspiration due to a decreased level of consciousness, predisposing to adverse outcomes in both short and long-term, independent of underlying neurologic injury. Future studies examining the underlying biologic mechanisms for delirium development, propagation, and resolution in stroke patients may be able to shed light on these questions. Presently what our findings reflect, is that delirium should be actively assessed for in stroke patients and should heighten the suspicion that these patients will suffer an adverse outcome not only during the current hospitalization but also that they are more likely to be readmitted and continue to experience poor

outcomes. Hence, delirium should serve as a vital sign prompting the bedside clinicians to carefully evaluate the patient and to look for factors that could be remedied.

The lack of association between delirium and mortality in hemorrhagic stroke patients is intriguing and does not follow the same pattern as observed for ischemic stroke subjects. We can postulate a few reasons for this discrepancy. Hemorrhagic stroke patients as a group constitute a sicker population with higher in-hospital and subsequent mortality as compared to ischemic stroke patients. For HS patients, we further analyzed the association between delirium and in-hospital mortality stratified across the Diagnosis Related Group (DRG) mortality risk categories (minor, moderate, severe and extreme). This association was similar to that observed for IS patients in the minor / moderate DRG category (ICH: aOR, CI: 1.36, 1.15–1.61, and SAH: aOR, CI: 1.58, 0.93–2.68), however it was different for the major and extreme categories (ICH: aOR, CI: 0.72, 0.64–0.81 and SAH: aOR, CI: 0.82, 0.66–1.02). It is therefore likely that the higher severity of illness predisposed the hemorrhagic stroke patients to mortality early in the hospital course and have reduced the probability of delirium assessment and diagnosis. As seen in the stratified analysis based on mortality risk, in hemorrhagic patients with low risk of mortality, delirium was associated with higher mortality. Additionally, patients in major and extreme risk category may have aggressive interventions planned and pursued during hospitalization, which could have accounted for not evaluating and documenting delirium in medical records. Daily monitoring of delirium at bedside using standardized tools will be able to provide information that can further clarify the link between delirium and mortality in hemorrhagic stroke patients.

Our study had limitations. We used an administrative database that may have resulted in an underestimation of delirium. Therefore, our estimates of delirium frequency should not be regarded as true delirium prevalence among hospitalized stroke patients. Also, due to lack of specific data on pre-morbid cognitive status and concurrent medications, we could not adjust for these factors in our main analyses. However, we conducted additional analyses while adjusting for concurrent diagnoses of mild cognitive impairment or dementia (MCID) and found that adjusted odds ratios and risk ratios for in-hospital and 30-day outcomes were similar to those reported from models without MCID as a covariate (S5 Table). The stroke patients represent a heterogeneous group. We classified them into hemorrhagic and ischemic subtypes but did not take into account the ischemic or hemorrhagic cerebral lesion location and disease specific severity measures. Finally, the results of our study may not be regarded as causal. Large sample size allowed us to control for several factors, though this accounts for measured confounding, it does not eliminate residual or unmeasured confounding. Consequently, the results of this study should be used primarily for generation of hypotheses.

In conclusion, the presence of delirium in stroke patients is associated with higher in-hospital mortality, longer length of hospital stay, unfavorable discharged disposition and a higher risk of 30-day readmissions. These data should stimulate more targeted tracking of this comorbidity in stroke patients and future prospective investigations into the clinical course both immediately and long-term for stroke patients who do and do not have delirium. Such information will likely prove helpful in early identification of delirium, initiation of effective management modalities, and prognostic information to help determine appropriate transition of care strategies to improve long-term patient outcomes and curtail overall healthcare burden.

## Supporting information

**S1 Table. ICD-9 codes for delirium.**
(DOCX)

**S2 Table. Descriptive and univariate analysis for all stroke types by delirium status (January 2010 to September 2015).**
(DOCX)

**S3 Table. In-hospital outcomes for stroke patient discharges with and without delirium by stroke-type.**
(DOCX)

**S4 Table. Rate of readmission and in-hospital outcomes among readmitted stroke patients with and without delirium by stroke-type.**
(DOCX)

**S5 Table.** Results of sensitivity analyses for association of concurrent mild cognitive impairment or dementia (MCID) with (a) delirium diagnosis and (b) in-hospital and 30-day outcomes among stroke discharges.
(DOCX)

## Author Contributions

**Conceptualization:** Farhaan S. Vahidy, Omar Rahman.

**Data curation:** Arvind B. Bambhroliya.

**Formal analysis:** Arvind B. Bambhroliya.

**Methodology:** Farhaan S. Vahidy, Arvind B. Bambhroliya.

**Project administration:** Farhaan S. Vahidy.

**Resources:** Farhaan S. Vahidy, Louise D. McCullough, Sean I. Savitz.

**Software:** Arvind B. Bambhroliya.

**Supervision:** Farhaan S. Vahidy.

**Visualization:** Farhaan S. Vahidy, Arvind B. Bambhroliya.

**Writing – original draft:** Farhaan S. Vahidy, Arvind B. Bambhroliya.

**Writing – review & editing:** Farhaan S. Vahidy, Arvind B. Bambhroliya, Jennifer R. Meeks, Omar Rahman, E. Wesley Ely, Arjen J. C. Slooter, Jon E. Tyson, Charles C. Miller, Louise D. McCullough, Sean I. Savitz, Babar Khan.

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
