## [Decision Letter · Decision Letter 0]

23 Sep 2019

PONE-D-19-25809

In-Hospital Outcomes and 30-day Readmission Rates among Ischemic and Hemorrhagic Stroke Patients with Delirium

PLOS ONE

Dear Dr Vahidy,

Thank you for submitting your manuscript to PLOS ONE. After careful consideration, we feel that it has merit but does not fully meet PLOS ONE’s publication criteria as it currently stands. Therefore, we invite you to submit a revised version of the manuscript that addresses the points raised during the review process.

We would appreciate receiving your revised manuscript by Nov 07 2019 11:59PM. To enhance the reproducibility of your results, we recommend that if applicable you deposit your laboratory protocols in protocols.io, where a protocol can be assigned its own identifier (DOI) such that it can be cited independently in the future. For instructions see: http://journals.plos.org/plosone/s/submission-guidelines#loc-laboratory-protocols

We look forward to receiving your revised manuscript.

Kind regards,

Aristeidis H. Katsanos, MD, PhD

Academic Editor

PLOS ONE

Journal Requirements:

3.

Thank you for stating the following in the Acknowledgments Section of your manuscript:

'We acknowledge the infrastructural support provided by the Biostatistics/ Epidemiology/Research Design (BERD) component of the Center for Clinical and Translational Sciences (CCTS) for this project. CCTS is mainly funded by a grant (UL1 TR000371) from the National Center for Advancing Translational Sciences (NCATS), awarded to University of Texas Health Science Center at Houston.'

'The author(s) received no specific funding for this work.'

Additional Editor Comments (if provided):

Reviewers' comments:

Reviewer's Responses to Questions

**Comments to the Author**

1. Is the manuscript technically sound, and do the data support the conclusions?

Reviewer #1: Yes

Reviewer #2: Partly

2. Has the statistical analysis been performed appropriately and rigorously? 

Reviewer #1: Yes

Reviewer #2: Yes

3. Have the authors made all data underlying the findings in their manuscript fully available?

Reviewer #1: Yes

Reviewer #2: Yes

4. Is the manuscript presented in an intelligible fashion and written in standard English?

Reviewer #1: Yes

Reviewer #2: Yes

5. Review Comments to the Author

Reviewer #1: Authors have performed a comprehensive study using Nationwide Claims data, assessing the prevalence of delirium and its association with outcomes among IS and hemorrhagic stroke patients. I have following comments:

1. Why did the authors select all-cause readmissions, instead of stroke related readmissions? I would suggest performing additional association of readmission 2/2 recurrent stroke or worsening of stroke symptoms with defined outcomes.

2. For patients with ICH, did the authors have specific data to differentiate the associations specifically for hemorrhagic conversion and parenchymal hematomas?

Reviewer #2: The present study assesses the consequences of in-hospital delirium in patients with stroke. The strengths of the study include the large patient population and the comprehensive statistical analysis. Major limitations are the utilisation of icd codes for the detection of diseases, comorbidities, and complications, the retrospective nature and the lack of data on the cognitive status of patients. Also, there was no adjustment made for possible comorbid dementia or mild cognitive impairment which are maybe the most important risk factors of delirium during hospitalisation (adjustment was made for ‘other’ neurological diseases in general). There is also no adjustment for chronic medications, for instance benzodiazepine use increases risk of delirium, too. All these very important limitations derive from the retrospective nature of the study based on diagnostic codes and are almost impossible to overcome. Consequently, the results of the study may be used primarily for hypotheses generation. All these have to be emphasised by the authors, as well as the need for further prospective studies with bedside assessment protocols.

6. PLOS authors have the option to publish the peer review history of their article (what does this mean?). If published, this will include your full peer review and any attached files.

Reviewer #1: Yes: Konark Malhotra

Reviewer #2: No

---

## [Author Response · Author response to Decision Letter 0]

7 Oct 2019

We appreciate the opportunity to revise and resubmit our manuscript. We also thank the editorial staff and the reviewers for their time in reviewing our work. We have made significant changes to the manuscript based on their comments and believe that our work has been strengthened by these changes. We have submitted a detailed response to each point raised by the academic editor and reviewers in the form of a rebuttal letter (Microsoft Word file) labeled as 'Response to Reviewers' in the submission system, a marked-up copy of our manuscript, highlighting changes made to the original version, labeled as ‘Revised Manuscript with Track Changes' and an unmarked version of our revised manuscript without tracked changes labeled as 'Manuscript'. We look forward to a reassessment of our revised manuscript.

---

## [Decision Letter · Decision Letter 1]

23 Oct 2019

PONE-D-19-25809R1

In-Hospital Outcomes and 30-day Readmission Rates among Ischemic and Hemorrhagic Stroke Patients with Delirium

PLOS ONE

Dear Dr Vahidy,

Thank you for submitting your manuscript to PLOS ONE. After careful consideration, we feel that it has merit but does not fully meet PLOS ONE’s publication criteria as it currently stands. Therefore, we invite you to submit a revised version of the manuscript that addresses the points raised during the review process.

We would appreciate receiving your revised manuscript by Dec 07 2019 11:59PM. To enhance the reproducibility of your results, we recommend that if applicable you deposit your laboratory protocols in protocols.io, where a protocol can be assigned its own identifier (DOI) such that it can be cited independently in the future. For instructions see: http://journals.plos.org/plosone/s/submission-guidelines#loc-laboratory-protocols

We look forward to receiving your revised manuscript.

Kind regards,

Aristeidis H. Katsanos, MD, PhD

Academic Editor

PLOS ONE

Reviewers' comments:

Reviewer's Responses to Questions

**Comments to the Author**

1. If the authors have adequately addressed your comments raised in a previous round of review and you feel that this manuscript is now acceptable for publication, you may indicate that here to bypass the “Comments to the Author” section, enter your conflict of interest statement in the “Confidential to Editor” section, and submit your "Accept" recommendation.

Reviewer #1: All comments have been addressed

Reviewer #2: All comments have been addressed

2. Is the manuscript technically sound, and do the data support the conclusions?

Reviewer #1: Yes

Reviewer #2: Yes

3. Has the statistical analysis been performed appropriately and rigorously? 

Reviewer #1: Yes

Reviewer #2: Yes

4. Have the authors made all data underlying the findings in their manuscript fully available?

Reviewer #1: Yes

Reviewer #2: Yes

5. Is the manuscript presented in an intelligible fashion and written in standard English?

Reviewer #1: Yes

Reviewer #2: Yes

6. Review Comments to the Author

Reviewer #1: All comments have been addressed by the authors.

No further modifications in the manuscript are needed.

Reviewer #2: Thank you for addressing the comments with such a comprehensive fashion. Please include the additional analysis conserving MCI and dementia (association and outcomes) as supporting information.

7. PLOS authors have the option to publish the peer review history of their article (what does this mean?). If published, this will include your full peer review and any attached files.

Reviewer #1: No

Reviewer #2: No

---

## [Author Response · Author response to Decision Letter 1]

28 Oct 2019

Dear Editor,

We appreciate the opportunity to revise and resubmit our manuscript revision #1. We have made changes to the manuscript as per the reviewers’ comments and responses are provided below. In addition to this rebuttal letter labeled as 'Response to Reviewers' in the submission system, we have provided a marked-up copy of our manuscript, highlighting changes made to the original version, labeled as ‘Revised Manuscript with Track Changes' and an unmarked version of our revised manuscript without tracked changes labeled as 'Manuscript'. We look forward to a quick reassessment of our revised manuscript. 

Regards,

Farhaan Vahidy.

Reviewers' Responses to Questions and Comments to the Author:

Reviewer #1: All comments have been addressed by the authors.

No further modifications in the manuscript are needed.

Response: We thank the reviewer for their time and effort.

Reviewer #2: Thank you for addressing the comments with such a comprehensive fashion. Please include the additional analysis conserving MCI and dementia (association and outcomes) as supporting information.

Response: We thank the reviewer for their time and effort. We have added the additional analysis that compares outcomes with and without MCID as a covariate (S5 Table). Corresponding revisions have been made to the manuscript as well. Please see pages 15 (lines 285-289, lines 377-379).

---

## [Editor Report · Decision Letter 2]

31 Oct 2019

In-Hospital Outcomes and 30-day Readmission Rates among Ischemic and Hemorrhagic Stroke Patients with Delirium

PONE-D-19-25809R2

Dear Dr. Vahidy,

We are pleased to inform you that your manuscript has been judged scientifically suitable for publication and will be formally accepted for publication once it complies with all outstanding technical requirements.

With kind regards,

Aristeidis H. Katsanos, MD, PhD

Academic Editor

PLOS ONE
---

## [Editor Report · Acceptance letter]

6 Nov 2019

PONE-D-19-25809R2 

In-Hospital Outcomes and 30-day Readmission Rates among Ischemic and Hemorrhagic Stroke Patients with Delirium 

Dear Dr. Vahidy:

I am pleased to inform you that your manuscript has been deemed suitable for publication in PLOS ONE. Congratulations! Your manuscript is now with our production department. 

With kind regards,

on behalf of

Dr. Aristeidis H. Katsanos 

Academic Editor

PLOS ONE